# Global warming reduces the carrying capacity of the tallest angiosperm species (*Eucalyptus regnans*)

Raphaël Trouvé [1] ✉, Patrick J. Baker[1], Mark J. Ducey [2], Andrew P. Robinson[3] & Craig R. Nitschke[1]

Rising temperatures and increased frequency and intensity of droughts and heat waves have affected tree mortality rates worldwide. Here, we investigate how these changes have affected the carrying capacity of mountain ash forests (*Eucalyptus regnans*), the world's tallest flowering plant and one of the most carbon-dense forests on earth. We analyze data from a large network of silvicultural experiments collected between 1947 and 2000 in southeastern Australia to identify trends in mortality rates and carrying capacity for the species, and to quantify how these changes relate to spatiotemporal variations in climate. We show that forests growing in the warmest and highest vapor pressure deficit conditions had the lowest carrying capacity, and this capacity further decreased with rising temperatures. Key findings indicate that a projected three °C increase in temperature by 2080 could reduce tree density and carbon stock in these forests by 24%, equivalent to losing 240,000 hectares of mature mountain ash forests. Trees that died were 0.62 times the size of living trees (*i.e.*, they were suppressed), with no detectable effect of climate on this ratio. We discuss the implications for forest conservation and management, and how reduced carrying capacity could undermine global forest restoration and carbon sequestration efforts.

Forests are one of the largest terrestrial carbon sinks in the world, absorbing about a third of our annual global emissions[1]. However, changes in climate threaten the ability of existing forests to remain net $CO_2$ absorbers[2]. This threat is being realized with increasing reports of tree mortality rates accelerating in many parts of the world[3–10] in response to rising temperature and drought intensity over the past century[11]. As climate warming continues, physiological models predict further drought-induced tree mortality to occur[6–8]. From an ecosystem perspective, the main concern is not changes in tree mortality rates per se, but changes in the forest's carrying capacity, which determines how much carbon a forest can hold. Changes in forest carrying capacity have the potential to turn forests from a carbon sink to a carbon source as they transition from higher to lower densities through increased tree mortality. Linking changes in mortality rates to changes in carrying capacity is essential if we want to predict where and how much forests will alter their carrying capacity under climate change.

In even-aged forests, carrying capacity can be described by the self-thinning line, which is the maximum number of trees of a given mean size that can be stocked per unit area[12]. Self-thinning is one of the few laws in plant population ecology[13] and results from an equilibrium between stand growth and density-dependent mortality in even-aged stands. As trees grow larger, they require more space and resources to survive, causing smaller trees to become increasingly suppressed and eventually die. Recent studies have advanced our mechanistic understanding of how tree shape, metabolism, and resource competition drive this self-thinning process[14–17]. Self-thinning lines have been shown

[1]The University of Melbourne, School of Agriculture, Food and Ecosystem Sciences, Richmond, Victoria, Australia. [2]University of New Hampshire, Department of Natural Resources and the Environment, 114 James Hall, Durham, NH, USA. [3]The University of Melbourne, CEBRA & School of BioSciences, Parkville, Victoria, Australia. ✉e-mail: raphael.trouve@unimelb.edu.au

to vary with species and site quality[18–23] but have traditionally been treated as time-invariant. Rapid changes in climatic conditions are challenging this assumption. Many regions are becoming more arid, with reduced soil water availability and increased atmospheric vapor-pressure deficits (VPD)[8,24]. In theory, this should result in reduced leaf area, a decrease in plot-level photosynthetic capacity, and a subsequent reduction in forest carrying capacity[25]. Despite this threat, few studies have explored how the self-thinning line changes over time in response to climatic variability. This is an important knowledge gap given the widespread changes in forest dynamics[26] and the need to detect and understand shifts in mortality rates and their drivers[27,28].

Also of interest is the partitioning of tree mortality within the stand. Climate change is unlikely to impact all trees equally[22,23,29,30]. However, it remains unclear which trees are most at risk. Studies in mixed-species forests have reported that large and tall trees are at greater risk[8,31–33], but many empirical studies in monocultures have consistently found higher sensitivity among smaller, suppressed trees[32,34–36].

Here, we build on recent advances in self-thinning modeling[37] to investigate changes in carrying capacity and their climate drivers in forests dominated by mountain ash (*Eucalyptus regnans*), the tallest flowering plant on Earth and one of the most carbon-dense forests[38–40]. Using long-term data spanning five decades (1947–2000), we quantify temporal changes in forest carrying capacity, their associations with climate conditions, and shifts in the relative size of dying trees and their potential climate drivers. Based on these findings, we project future changes in forest carbon stocks under climate warming scenarios.

## Results

Both temperature and mountain ash mortality rates have increased over time (Fig. 1). Temperatures in the region increased by 0.71°C between 1947 and 2000, with some noticeable variability from year-to-year. Over the same period, mortality rates for *E. regnans* (adjusted for tree density and tree size) increased by 33%.

Statistical modeling of the effect of Mean Annual Temperature (MAT) on self-thinning showed that higher temperatures were associated with lower carrying capacity. The effect size was consistent whether the impact of temperature was modeled across space using static self-thinning allometries (Fig. 2 and Table 1: Eq. (1)) or in time

using dynamic self-thinning allocations (Table 1: Eq. (2)) or a mix of both (Table 1: Eq. (3)). Carrying capacity was lowest for stands growing in the warmest conditions and further decreased with rising temperatures within each plot. On average, each additional one °C increase in temperature was associated with a 24.6% increase in mortality rates and a 9% reduction in the carrying capacity of mountain ash stands (Fig. 3 and Table 1). Based on these relationships, the model predicts that a three °C increase in MAT will result in a 24% decline in tree density and carbon content in these forests. The CSIRO RCP8.5 scenario projects this region will experience those temperatures by ~2080.

Trees that died were, on average, 62% the size of live trees (*i.e.*, a *k* factor of 0.62, see Fig. 4). We found no evidence for the effect of temperature, precipitation, Annual Heat Moisture Index (AHMI, an aridity index often used to model tree growth in the region[41]), or year on the relative size of trees that died (see Supplementary Note 4).

## Discussion

Human activities have caused the Earth's temperature to rise nearly 1.5°C since the beginning of the Industrial Age, with this trend expected to continue in the future[11]. Here, we predicted that a three °C increase in temperature will decrease the carrying capacity of *E. regnans* by 24%. Trees that died were on average 0.62 times smaller than the surviving trees (*i.e.*, they were suppressed trees), with no detectable effect of climate on this pattern.

Despite numerous studies on spatial variation in self-thinning[21,22,42–49], we are only aware of four studies exploring temporal variation. Three from Northern Europe and Canada reported increased carrying capacities over time[50–52], while a fourth study spanning from Spain to Sweden found no significant temporal changes[53]. This geographical pattern suggests that the trend in carrying capacity may depend on regional climate constraints. In temperature-limited systems, increased carrying capacity may be due to longer growing seasons and higher $CO_2$ concentrations[54] or perhaps, in the case of European forests, to increased nitrogen deposition[55,56]. However, forests in warmer, seasonally dry climates face different limitations. In these systems, such as Mediterranean and temperate forests, including our mountain ash forests of southeast Australia, tall trees with high leaf area become increasingly vulnerable to atmospheric water demand as temperatures rise[8], limiting the

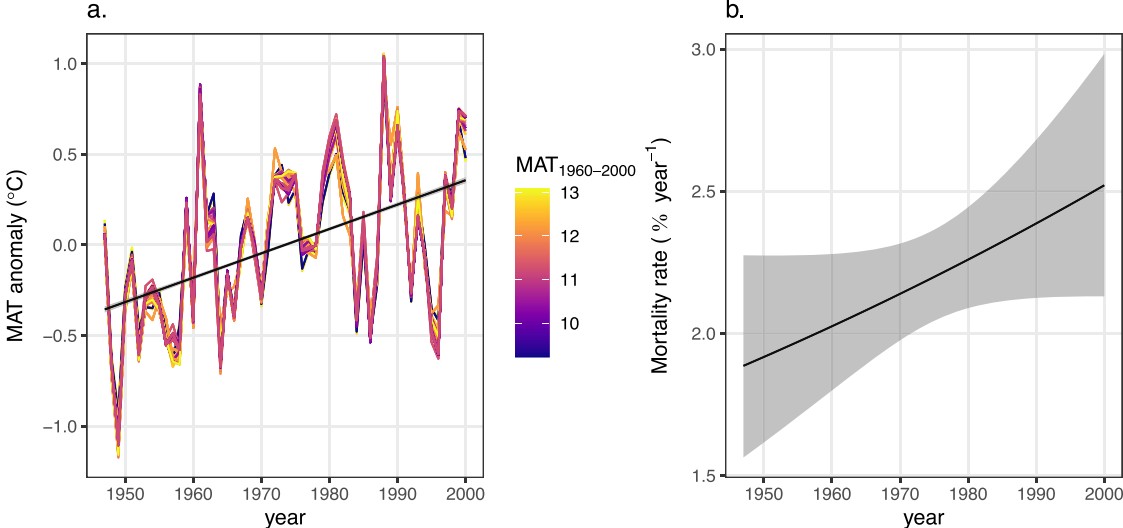

**Fig. 1 | Temporal trends in temperature and mortality rates. a** Mean annual temperature (MAT) anomalies vs. year for each plot. Colored lines represent individual plots, with colors indicating each plot's average MAT over the 1960–2000 period. Anomalies show the difference between each year's temperature and the plot's mean temperature over 1960–2000, illustrating the temporal warming trends. **b** Modeled mean mortality rates for mountain ash vs. year. The solid line shows the mean fitted value and the shaded area shows the 95% CI.

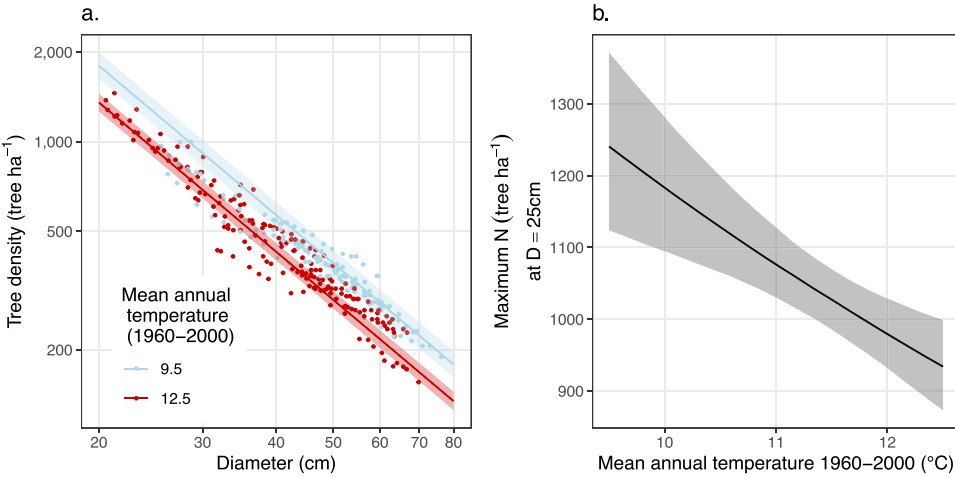

**Fig. 2 | The influence of temperature on self-thinning allometry. a** Effect of $MAT_{1960-2000}$ on maximum stand stocking per ha. Mean model predictions from Eq. (1) are shown at 9.5 °C (blue line) and 12.5 °C (red line), representing the approximate temperature range in our dataset and demonstrating the expected impact of a 3 °C warming on maximum stocking. Data points are colored by whether their $MAT_{1960-2000}$ is below (blue) or above (red) the dataset's mean temperature of 11 °C. **b** Effect of $MAT_{1960-2000}$ on maximum stand stocking at D = 25cm (*i.e.*, stand density index) showing a 9% decrease in carrying capacity for each additional °C increase in temperature. Each dot represents one plot measurement. The shaded areas show 95% credible intervals.

forests' ability to support high biomass levels. Our findings support this mechanism: as the climate warmed, mountain ash forests sustained fewer trees. While the strong correlation between temperature and VPD in our study region (Supplementary Fig. 2) makes their individual effects difficult to disentangle, this relationship suggest that warming effects may be mediated through rising VPD, an important driver of drought-induced plant mortality[57,58].

Our results offer insights into the commonly used technique of space-for-time substitution. By using long-term experiments covering wide climatic gradients and leveraging recent self-thinning methods with improved temporal resolution, we were able to disentangle spatial and temporal variations in carrying capacity and identify climate as a causal factor. We found that a one degree temperature increase among sites (*i.e.*, in space) and a one degree increase within sites (*i.e.*, in time) had similar effect sizes. This provides compelling evidence for a projected decrease in carrying capacity due to climate warming and supports the often assumed but rarely tested space-for-time substitution[59,60] for our case study.

Changes in forest carrying capacity have knock-on effects on many ecosystem services. Perhaps the most direct consequence is a drop in forest carbon stock. Here, we can distinguish between temporary and more permanent changes in the self-thinning line (Fig. 5). For example, during a multi-year heatwave or drought (Fig. 5a.), the forest's carrying capacity will temporarily drop (*i.e.*, the forest becomes a carbon source) before recovering its initial trajectory as the conditions ease (*i.e.*, the forest is once again a carbon sink). With climate warming, the drop in carrying capacity is likely to be more permanent (Fig. 5b).

This predicted 24% drop in carrying capacity at the stand scale does not include the impact of rising risk posed by changes to fire regimes. Warmer temperatures and increasing aridity are likely to drive more frequent fires[61], resulting in stands with lower carbon density[62]. Additionally, regeneration failures may cause less carbon-dense species to replace parts of the landscape[63–65], leading to a further reduction in carbon stocks beyond this predicted 24% decrease. *Eucalyptus regnans* forests are among the most carbon-dense on Earth, storing 450–819 tonnes of carbon per ha in live trees[66,67]. Extrapolating the 24% carrying capacity drop to the million hectares of tall-open wet forests in Victoria and Tasmania would mean, at minimum (using the 450 tonnes carbon per ha

value–representative of a typical stands–rather than the 819 tonnes carbon per ha value representative of old growth forests), a loss of around 108 million tonnes of carbon stock by 2080 compared to current conditions. For context, this would be equivalent to one million people driving 10,000 km per year for 75 years. The average car emission in Australia is -146 g carbon per km[68]. This predicted carbon loss is also equivalent to the loss of 240,000 ha of mature mountain ash forests. The tall-open wet forests of the Central Highlands provide essential water supply for Melbourne's five million inhabitants. Recent studies highlighted that streamflow behavior and water yield were sensitive to self-thinning parameters[69,70]. This means that any alterations to the self-thinning line are likely to have implications for Melbourne's water supply.

Trees that died were consistently smaller than their neighbors (*i.e.*, they were suppressed), averaging 62% of their diameter (Fig. 4). This size ratio was stable across all tested climatic gradients. While several studies report greater drought-related mortality among larger trees[31–33,71], others, including ours, find higher vulnerability among suppressed individuals[32,34–36]. Reconciling these patterns requires accounting for relative size within a stand (*i.e.*, social or canopy status), absolute size, and potential confounders such as stand structure and species composition. In mixed-species forests, the apparent vulnerability of large trees can stem from drought-sensitive taxa, particularly fast-growing pioneer species, dominating the upper canopy[72], but see ref. 73 for an alternative view. Size effects may also be driven or amplified by indirect mortality agents, for example bark beetles which preferentially attack large pines during droughts[74]. By contrast, our monospecific, even-aged stands lack these confounders, and mortality concentrates among suppressed trees, consistent with self-thinning dynamics. The vulnerability of suppressed trees stems from physiological and morphological constraints: reduced light interception, smaller carbon reserves, and limited access to deep soil water due to shallow root systems[75–78].

Changes in carrying capacity pose a significant challenge to global-scale tree restoration initiatives, which have been proposed as a key mitigation strategy to reduce atmospheric $CO_2$[79]. Despite their potential, such initiatives may not deliver their expected outcomes due to carrying capacity changes that are currently unaccounted for. The density-dependence nature of temperature and drought related mortality[25] suggests that reducing stand density can improve forest

**Table 1 | Parameter summary for Eqs. (1), (2), and (3)**

| Parameter | Predictor | Mean | 95% CI Min | 95% CI Max |
|---|---|---|---|---|
| Static allometry (Eq. (1)) | | | | |
| $\alpha_O$ | Intercept STL | 12.29 | 12.18 | 12.39 |
| $\alpha_1$ | log(D) | − 1.67 | − 1.69 | − 1.64 |
| $\alpha_2$ | $MAT_{1960-2000_j}$ | − 0.09 | − 0.14 | − 0.05 |
| $\sigma_{\alpha_0}$ | Plot-level SD | 0.12 | 0.10 | 0.16 |
| $\sigma$ | Residual SD | 0.06 | 0.05 | 0.06 |
| Mortality model (Eq. (2)) | | | | |
| $\beta_O$ | Intercept | − 29.38 | − 32.48 | − 26.36 |
| $\beta_1$ | log(D) | 2.92 | 2.51 | 3.33 |
| $\beta_2$ | log(N) | 2.38 | 2.11 | 2.66 |
| $\beta_3$ | $MAT_{1960-2000_j}$ | 0.15 | − 0.05 | 0.36 |
| $\beta_4$ | $MAT_{anomaly_{ij}}$ | 0.28 | 0.09 | 0.47 |
| $\sigma_{\beta_0}$ | Plot-level SD | 0.84 | 0.69 | 1.03 |
| $\theta$ | Shape NB | 9.39 | 6.69 | 13.04 |
| Intercept | Intercept STL | 12.55 | 12.25 | 12.88 |
| Slope | Slope STL | − 1.65 | − 1.72 | − 1.58 |
| $\frac{\beta_3}{-\beta_2}$ | $MAT_{1960-2000_j}$ | − 0.06 | − 0.15 | 0.02 |
| $\frac{\beta_4}{-\beta_2}$ | $MAT_{anomaly_{ij}}$ | − 0.12 | − 0.20 | − 0.04 |
| Mortality model (Eq. (3)) | | | | |
| $\beta_O$ | Intercept | − 29.61 | − 32.67 | − 26.66 |
| $\beta_1$ | log(D) | 2.95 | 2.56 | 3.35 |
| $\beta_2$ | log(N) | 2.40 | 2.13 | 2.68 |
| $\beta_5$ | $MAT_{ij}$ | 0.22 | 0.09 | 0.36 |
| $\sigma_{\beta_0}$ | Plot-level SD | 0.84 | 0.69 | 1.03 |
| $\theta$ | Shape NB | 9.42 | 6.71 | 13.12 |
| Intercept | Intercept STL | 12.55 | 12.26 | 12.87 |
| Slope | Slope STL | − 1.65 | − 1.72 | − 1.58 |
| $\frac{\beta_5}{-\beta_2}$ | $MAT_{ij}$ | − 0.09 | − 0.15 | − 0.04 |

The effect size of MAT on the intercept of the self-thinning line (STL) is estimated by $\alpha_2$, $\frac{\beta_3}{-\beta_2}$, $\frac{\beta_4}{-\beta_2}$, and $\frac{\beta_5}{-\beta_2}$.

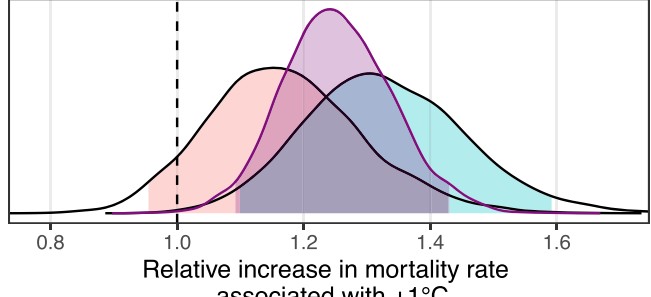

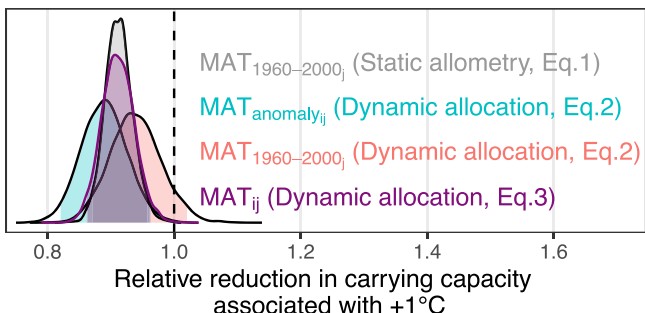

**Fig. 3 | Estimated impact of a one °C increase in temperature on mortality rate and carrying capacity.** The top figure is the posterior distributions for $e^{\beta_3}$, $e^{\beta_4}$, and $e^{\beta_5}$ (multiplicative effect associated with a one °C increase on mortality rate). The bottom figure is the posterior distributions for $e^{\frac{\beta_3}{-\beta_2}}$, $e^{\frac{\beta_3}{-\beta_2}}$, $e^{\frac{\beta_4}{-\beta_2}}$, and $e^{\frac{\beta_5}{-\beta_2}}$ (multiplicative effect associated with a one °C increase on maximum tree density). On average, carrying capacity dropped by 9% for each additional °C increase in temperature.

resilience to climate change. The potential of forest thinning to mitigate drought stress has been highlighted in many studies[6,80,81] although there is still debate on the consistency of this effect[82]. Our conceptual framework provides additional insights into this issue: as the self-thinning line drops, stands that were on the self-thinning line become overstocked and suffer excess mortality[83], while stands with sufficiently low densities avoid being caught on the wrong side of the self-thinning line and fare better. We suggest using climate sensitive self-thinning diagrams and forecasted trajectories (Fig. 5) to guide density management recommendations in a changing environment. This will allow for the implementation of climate-smart forest management that promotes resilience in areas of the landscape where stands are vulnerable to climate change[84]. These results call for a broader investigation into the dynamic nature of carrying capacity, with significant implications for ecosystem services and the development of mitigation strategies.

## Methods
### Case study species and area
This study focuses on the tall-open wet forests of the Central Highlands of Victoria, Australia. These forests are dominated by mountain ash (*E. regnans*), the tallest angiosperm (*i.e.*, flowering plant) in the world. Living individuals reach heights of 100 meters[39]. Their great height and typical occurrence in mono-specific stands allow the forests to accumulate extraordinary amounts of above-ground carbon per unit area [> 450–819 tonnes carbon per ha[66,67]]. Tall-open wet forests cover an area of ~1,000,000 ha spread across Tasmania and Victoria. Here, we focus specifically on the Central Highlands of Victoria, a mid-mountainous range located 80 km east of Melbourne, covering around 400,000 ha, including ~30% of tall-open wet forests. In the Central Highlands, the tall-open wet forests occur from 200–1100 m above sea level with mean annual rainfall of 1000–1600 mm.

### Field data
The data in this study come from a database of long-term silvicultural experiments in even-aged stands from Victoria's Central Highlands. Most of the experiments were started in the 1960s by the Victorian Forestry Commission to study the effect of thinning on growth and yield. Each experimental site had several permanent plots and included a range of thinning treatments, plus controls. Mean plot size was 2600 m² (ranging from 400 m² to 8000 m²). Individual trees within each plots were measured for diameter at breast height (DBH, measured at 1.3 m) and status (alive, dead) every 2–3 years until the late 1990s or early 2000s. The data was originally collected by the Forestry Commission. When the Forestry Commission was dis-established in 1983, the Victorian Department of Energy, Environment and Climate Action (DEECA) and its precursors managed and curated the data.

We selected pure *E. regnans* plots from the database based on the following criteria[37]: more than 80% of the basal area was from *E. regnans*, plots were sufficiently large (≥400 m²) with sufficiently long intercensus intervals (≥0.5 year) and growth (ΔD≥0.1 cm year⁻¹). We excluded plots that experienced fire, psyllid infestation, or storm damage. For each plot and each measurement, we calculated the

quadratic mean diameter ($D$ in cm) and tree density (trees ha$^{-1}$). For each pair of successive measurements on the same plot, we counted the number of dead trees ($\Delta N$, in trees) and calculated the net increase in quadratic diameter ($\Delta D$, in cm) during that period. In total, we had 1302 measurements from 112 plots, including 328 measurements from 40 unthinned control plots that showed no evidence of having experienced disturbance over their measurement period. See Supplementary Fig. 1 for the plot location and Supplementary Table 1 for the data summary.

## Modeling variation in self-thinning associated with variation in climatic conditions

The self-thinning line describes the maximum number of trees of a given mean size that can be stocked per unit area. Since the maximum stocking level described by the self-thinning line depends on the amount of resources available, we expect it to decrease as climatic conditions become less favorable, which in these forests translates to being warmer and drier.

Self-thinning modeling methods fall into two categories: traditional methods based on static size-density allometry, and more recent methods based on the dynamic allocation between stand growth and mortality[37]. Static allometry methods are straightforward to use and well suited to detecting spatial variations in self-thinning and their environmental drivers. However, because static allometries integrate the impact of past conditions over the entire stand's lifespan, they are poorly suited to quantifying temporal changes in self-thinning. Dynamic allocation methods have better temporal resolution and can detect both spatial and temporal variations in self-thinning; however, they typically require more data than static allometry methods.

**Spatial variation in the self-thinning line based on static size-density allometry.** We first analysed spatial variation in the static size-density allometry associated with variation in mean climatic conditions among plots. We used linear mixed-effects models on selected control plots[37] to estimate the self-thinning parameters. We accounted for repeated measurements per plot using plot-level random effects. The linear mixed-effects model that we used was:

$$\log(N_{ij}) \sim \mathcal{N}(\mu_{ij}, \sigma)$$
$$\mu_{ij} = \alpha_{0j} + \alpha_1 \log(D_{ij}) + \alpha_2 \text{MAT}_{1960\text{-}2000_j} \quad (1)$$
$$\alpha_{0j} \sim \mathcal{N}(\alpha_0, \sigma_{\alpha_0})$$

where $\log(N_{ij})$ is the natural logarithm of tree density for observation $i$ in plot $j$, which is modeled as being sampled from a Normal distribution with mean $\mu_{ij}$ and standard deviation $\sigma$; $D_{ij}$ is the quadratic mean diameter associated with observation $i$ in plot $j$; and $\text{MAT}_{1960\text{-}2000_j}$ is the average mean annual temperature of plot $j$ over the 1960 to 2000 period. Climate variables (temperature and precipitation) were extracted from 1 km resolution Australian Gridded Climate Data[85]. Vapor pressure deficit data came from a separate 250 m resolution

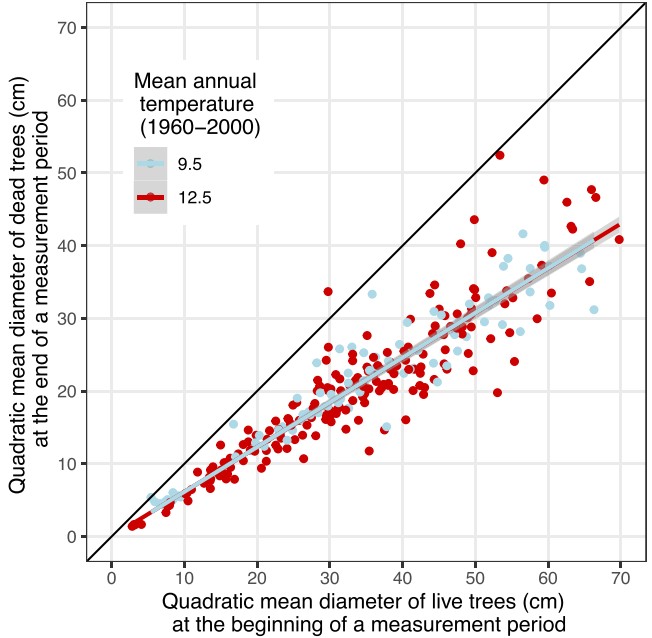

**Fig. 4 | Relative size of trees that died during a mortality event.** Each dot represents the quadratic mean diameter of dead trees at the end of a measurement period vs. quadratic mean diameter (of live trees) at the beginning of the period. The figure only includes mortality events that had more than three dead trees recorded to reduce dispersion. The solid black line represents the 1:1 line (*i.e.*, the DBH of dead trees at the end of a measurement period equals the DBH of live trees at the beginning of the period). For visualization purposes, plots were colored according to their MAT$_{1960–2000}$ whether they fall above or below 11°C. The red and blue lines show mean predictions from linear regression fits to the data. The slope equals 0.62. The shaded areas show 95% CI. We found no significant effect of temperature, precipitation, AHMI, or date on the relative size of dead trees (see Supplementary Table 4).

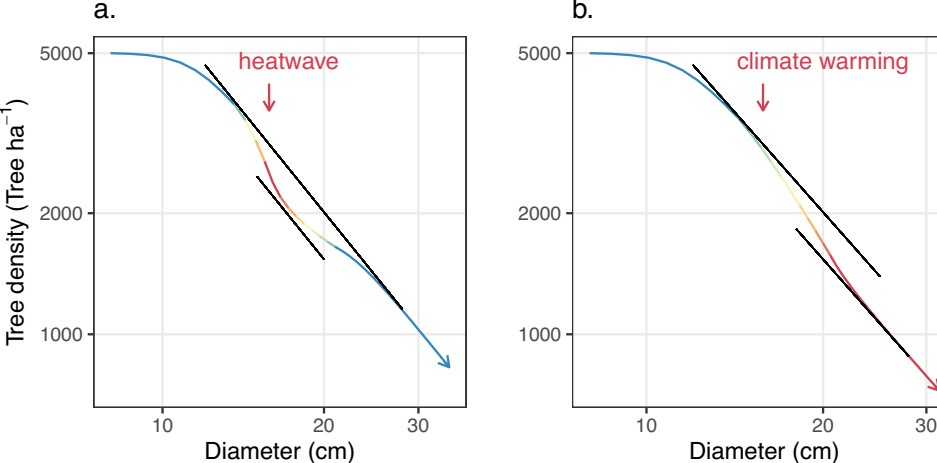

**Fig. 5 | Predicted effect of a transient vs. permanent three °C increase in temperature on the self-thinning trajectory. a** Heatwaves cause a temporary reduction in the self-thinning line. As the conditions ease, the stand recovers its original trajectory. **b** Climate change might cause a more permanent reduction in the self-thinning line.

dataset that only covers the period after 1980[86]. $\alpha_0$ is the population-level intercept of the self-thinning line, $\alpha_1$ is the slope of the self-thinning line, and $\alpha_2$ represents the impact of $MAT_{1960\text{-}2000_j}$ on the self-thinning intercept (*i.e.*, variation among plots in the intercept of the self-thinning line due to differences in mean annual temperature). $\sigma_{\alpha_0}$ is the standard deviation associated with the plot-level random effect (*i.e.*, variation among plots that is not associated with differences in mean annual temperature) and $\alpha_{0j}$ is a plot-specific intercept.

We considered four climatic predictors that might impact the self-thinning line: mean annual temperature (MAT), annual precipitation ($P$), vapor pressure deficit (VPD), and the annual heat moisture index (AHMI). AHMI combines MAT and precipitation in a simple ratio: $AHMI = \frac{MAT+10}{P/1000}$. This metric, which is the inverse of the classic Martonne aridity index[87], has proven valuable for modeling tree growth in the region[41]. Vapor pressure deficit, which measures the difference between actual and saturated air moisture, has been linked to drought-induced plant mortality[58]. Before fitting the models, we transformed the climate variables to simplify interpretation: MAT and VPD were centered, while P and AHMI were both centered and scaled. This ensured the intercept reflects the average population response.

Among the four climate predictors (parameters shown in Supplementary Table 2), only MAT and VPD had significant effects on the self-thinning line. The MAT model performed best in leave-one-plot-out cross-validation, with an RMSE of 0.131, compared to 0.136 for VPD, 0.154 for precipitation, and 0.152 for AHMI (see Supplementary Table 3). Given the strong correlation between MAT and VPD in our dataset (R = 0.96, see Supplementary Fig. 2), their similar performance was expected. However, due to VPD data availability (starting only from 1980 versus records extending back to 1900 for the other variables) and the lack of reliable VPD projections for future climate scenarios, we focused subsequent analyses on MAT while acknowledging that its effects may be mediated through increased atmospheric water demand (*i.e.*, rising VPD).

**Spatiotemporal variation in the self-thinning line based on growth–mortality allocation.** We then used recent advances in self-thinning modeling to quantify both spatial and temporal changes in the self-thinning line linked to variation in climatic conditions.

We first used a Generalized Linear Mixed Model (GLMM) with a Negative Binomial family and a log-link to calibrate a mortality model[37]. We accounted for repeated measurements per plot using plot-level random effects. The mortality model that we used was:

$$
\begin{aligned}
\Delta N_{ij} &\sim NB(\mu_{ij}, \theta) \\
\log(\mu_{ij}) &= \beta_{0ij} + \beta_1 \log(D_{ij}) + \beta_2 \log(N_{ij}) + \log(\Delta D_{ij}) + \log(N_{raw_{ij}}) \\
\beta_{0ij} &= \beta_{0j} + \beta_3 MAT_{1960\text{-}2000_j} + \beta_4 MAT_{anomaly_{ij}} \\
\beta_{0j} &\sim \mathcal{N}(\beta_0, \sigma)
\end{aligned}
\tag{2}
$$

where $\Delta N_{ij}$ is mortality count for observation $i$ in plot $j$ and $\Delta D_{ij}$ are observed values for individual inventory $i$ in plot $j$, $\mu_{ij}$ is the mean modeled mortality count and $\beta_{0ij}$, $\beta_1$, $\beta_2$, $\beta_3$, and $\beta_4$ are parameters to be estimated. $\beta_{0j}$ is a plot-level intercept and $\theta$ is the shape parameter of the negative binomial distribution. $N_{raw_{ij}}$ is an offset for the number of trees observed in plot $j$ at time $i$ that is used to convert raw mortality counts into mortality rates.

For the dynamic self-thinning allocation model, we tried to disentangle the spatial and temporal impact of climatic conditions on carrying capacity by using a combination of variable transformations. To model the impact of spatial variation in climatic conditions among plots, we used average mean annual temperature of plot $j$ over the 1960 to 2000 period ($MAT_{1960\text{-}2000_j}$). $MAT_{1960\text{-}2000_j}$ is the same predictor that was used in Eq. (1) to estimate spatial variation in the static self-thinning allometry. To model the impact of temporal variation in climatic conditions within each plot, we calculated climatic

anomalies ($MAT_{anomaly_{ij}}$) by measuring the difference between the plot's conditions in each intercensus period ($MAT_{ij}$) and its average conditions ($MAT_{1960\text{-}2000_j}$). Finally, to test whether plots in warmer sites were more likely to respond to climate warming than plots in colder sites, we included an interaction term between $MAT_{1960\text{-}2000_j}$ and $MAT_{anomaly_{ij}}$. Since this interaction was not significant and did not improve model fit, it was not included in the final model. However, this preliminary analysis revealed that $MAT_{1960\text{-}2000_j}$ and $MAT_{anomaly_{ij}}$ had similar effect sizes. To simplify the model and allow for more robust inference, we also fitted a simplified mortality model that replaced the $MAT_{1960\text{-}2000_j}$ and $MAT_{anomaly_{ij}}$ predictors with a composite predictor that represented the mean annual temperature experienced by a plot for a specific intercensus period (*i.e.*, $MAT_{ij} = MAT_{1960\text{-}2000_j} + MAT_{anomaly_{ij}}$). The simplified mortality model was:

$$
\begin{aligned}
\Delta N_{ij} &\sim NB(\mu_{ij}, \theta) \\
\log(\mu_{ij}) &= \beta_{0ij} + \beta_1 \log(D_{ij}) + \beta_2 \log(N_{ij}) + \log(\Delta D_{ij}) + \log(N_{raw_{ij}}) \\
\beta_{0ij} &= \beta_{0j} + \beta_5 MAT_{ij} \\
\beta_{0j} &\sim \mathcal{N}(\beta_0, \sigma)
\end{aligned}
\tag{3}
$$

where $\beta_5$ represents the impact of $MAT_{ij}$ on mortality.

In models without climatic predictors, the intercept and slope of the self-thinning line are derived from the mortality models as follows[37]:

$$
intercept = \frac{\beta_0}{-\beta_2} + \frac{\log(\beta_2) - \log(\beta_1 + 1)}{-\beta_2}
\tag{4}
$$

$$
slope = \frac{1 + \beta_1}{-\beta_2}
\tag{5}
$$

The negative ratio of $\beta_0$ and $\beta_2$ represents the intercept (the terms in the second part of the equation are logged and are therefore mostly negligible). The slope of the self-thinning line is calculated from $\beta_1$ and $\beta_2$. The impact of climatic predictors on the self-thinning line is best understood as a modifier of $\beta_0$. The effect size of $MAT_{1960\text{-}2000_j}$, $MAT_{anomaly_{ij}}$, and $MAT_{ij}$ on the intercept of the self-thinning line is thus $\frac{\beta_3}{-\beta_2}$, $\frac{\beta_4}{-\beta_2}$, and $\frac{\beta_5}{-\beta_2}$, respectively. The relative reduction in carrying capacity associated with a one °C increase in temperature can be calculated by computing the exponent of these ratios. For example, if $\beta_5 = -0.22$ and $\beta_2 = 2.4$, then the additive impact of $MAT_{ij}$ on the intercept of the self-thinning line is $\frac{\beta_5}{-\beta_2} = -0.09$ and the multiplicative impact of $MAT_{ij}$ on the carrying capacity is $e^{-0.09} = 0.91$. In this example, for each degree increase in $MAT_{ij}$, the carrying capacity (*i.e.*, maximum tree density per ha for a stand of a given quadratic mean diameter) of the forest drops by 9%. For a three °C increase in $MAT_{ij}$, the carrying capacity of the forest is multiplied by $e^{-0.09 \times 3} = 0.76$, *i.e.*, a 24% drop.

We fit all models using the 'brms' package[88] in R[89] with weakly informative priors[90]. Uncertainty in the derived parameters was propagated by sampling from the joint posterior distribution (*i.e.*, MCMC samples) of model parameters to obtain a posterior distribution for the derived parameters. We then summarized the posterior distribution by computing the mean and 95% credible intervals for each parameter.

## Model evaluation
We assessed the accuracy of the final models using three common metrics: the coefficient of determination ($R^2$, Eq.(6)), root mean squared error (RMSE, Eq.(7)), and bias (Eq.(8)), all calculated using a leave-one-plot-out cross-validation procedure.

$$
R^2 = 1 - \frac{\sum (y_i - \hat{y}_i)^2}{\sum (y_i - \bar{y})^2}
\tag{6}
$$

$$\text{RMSE} = \sqrt{\frac{\sum (y_i - \widehat{y_i})^2}{n}} \qquad (7)$$

$$\text{BIAS} = \frac{\sum y_i - \widehat{y_i}}{n} \qquad (8)$$

In these equations, $y_i$ is the observed value (*i.e.*, log($N$) for Eq. (1) and mortality count and mortality rate for Eqs. (2) and (3)) for inventory $i$ and $\widehat{y_i}$ is the model's prediction. $\overline{y}$ is the average observed value in the dataset.

To calculate these metrics, we used leave-one-plot-out cross-validation. This involved training the model on all data except one plot, predicting for the left-out plot, and repeating the process for each plot. We then used these out-of-sample predictions to compute the goodness-of-fit metrics.

On out-of-sample data, the static self-thinning allometry (Eq. (1)) had a $R^2$ of 0.973, RMSE of 0.131, and bias of -0.010 in terms of log($N$) (see Supplementary Table 3). The final mortality model (Eq. (3)) had a $R^2$ of 0.984, RMSE of 58.9, and bias of -4.5 in terms of mortality count per ha and a $R^2$ of 0.719, RMSE of 0.019, and bias of -0.0049 in terms of mortality rate (see Supplementary Table 3).

### Forecasting the impact of climate warming on carrying capacity

To assess the impact of climate warming on forest carrying capacity, we computed the self-thinning line using a future climate that is three °C warmer than the current levels. This corresponds to the RCP8.5 scenario developed by CSIRO for the region[41]. We compared the effects of a transient (*e.g.*, 10 year heatwave) versus a permanent (*e.g.*, climate warming) three °C temperature increase on stand survival trajectory.

### Impact of relative tree size on mortality

We tested the potential impact of climatic factors and year on the relative size of trees that die. In each successive pair of measurements on the same plot, we recorded the quadratic mean diameter of the live trees at the beginning of the period and the quadratic mean diameter of the trees that died by the end. The ratio of these two measurements is often referred to as the $k$ factor, where $k = \frac{D_{dead}}{D}$.

We used linear mixed models to test whether the $k$ factor was influenced by climatic factors (MAT, precipitation, AHMI), temporal trends (*i.e.*, calendar year), and stand dominant height (Ho). We tested for dominant height as taller trees might be more susceptible to drought due to increased hydraulic constraints[91]. We used a plot-level random effect to account for the fact that measurement from the same plots are not independent. The model structure is shown in Eq. S7 in Supplementary Note 4. Parameter estimates for the fitted models are available in Supplementary Table 4. None of the tested predictors (climate variables, time trends, or dominant height) were significant, indicating that suppressed trees were consistently more likely to die than other trees across all conditions.

### Reporting summary

Further information on research design is available in the Nature Portfolio Reporting Summary linked to this article.

## Data availability

The data used in this study are available in the Zenodo repository https://doi.org/10.5281/zenodo.15686632[92].

## Code availability

The R code used in this study is available on the Zenodo repository https://doi.org/10.5281/zenodo.15686632[92].

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

## Acknowledgements

This work was supported by the Australian Research Council (ARC), through an ARC Discovery Project (DP220103711 to C.R.N., R.T., A.P.R., P.J.B., and M.J.D.). We appreciate the efforts of successive staff from the Victorian Forestry Commission, VicForests, and now DEECA in maintaining and providing access to the long-term permanent sample plot database used in this study.

## Author contributions

R.T., C.R.N., and P.J.B. conceived and designed the study. A.P.R., C.R.N., M.J.D., P.J.B., and R.T. acquired the funding. R.T. analyzed the data with assistance from A.P.R., C.R.N., M.J.D., and P.J.B. R.T. drafted the manuscript. All authors contributed to revising the manuscript and approved the final manuscript.

## Competing interests

The authors declare no competing interests.
