## [Transparent Peer Review file · Nature Communications]

Global warming reduces the carrying capacity of the tallest angiosperm species (*Eucalyptus regnans*)

Corresponding Author: Dr Raphael Trouve

Version 0:

Reviewer comments:

Reviewer #1

(Remarks to the Author)

This manuscript examines how spatial and temporal differences in climate influence the mortality of *Eucalyptus regnans* forests in Australia. Self-thinning models are fit to tree measurements from plots with many repeated measurements. Importantly, the approach quantifies the effects of climate. An understanding of mortality dynamics is critical for understanding the influence of climate change on forest carbon stocks, and this study shows how such information can be obtained. The manuscript is therefore very interesting and is likely to be of great value to many in this field. It is also well written and I did not note any further suggestions or comments regarding the main text or supplementary information.

(Remarks on code availability)

When I ran the R code, I had an error message "Error: Please install the 'cmdstan' package." But when I tried to install it, I had an error message "Error in FUN(X[[i]], ...) : there is no package called 'cmdstan'".

Reviewer #2

(Remarks to the Author)

The manuscript "Global warming reduces carrying capacity for tallest angiosperm" investigated how drought and heat waves have impacted the carrying capacity of mountain ash forests (*Eucalyptus regnans*), by analyzing silvicultural data from 1947 to 2000 in southeastern Australia. Authors found that warmer conditions and projected temperature increases could reduce tree density and carbon stocks by 24%, with significant implications for forest conservation and global carbon sequestration efforts.

The paper links self-thinning and mortality rates to predict potential carrying capacity dynamics under climate change, providing critical implications for the restoration and management of global forests for carbon sequestration. It is clearly written and includes simple, direct figures, making it an enjoyable read. In my personal view, this manuscript is already very strong. However, I have one suggestion: could the authors make the paragraphs more concise by avoiding the use of single-sentence or two-sentence paragraphs?

(Remarks on code availability)

Reviewer #3

(Remarks to the Author)

The study investigates the stand density/ tree number relationship for Australian trees that potentially grow very high. The authors find that tree mortality is higher and stand density is smaller at warmer sites which are assumed to be dryer too. These results are used to interpolate the biomass carrying capacity into a future that is 30C warmer.

Although the study findings originate from a specific region, I think that they may likely be of general interest and applicable to similar systems. The results are clearly presented and the message is interesting, although neither globally valid nor unique. It is a pleasure to read.

I have, however, some concerns about the discussion of and conclusions from results. Although most of the relevant literature is considered, some papers are clearly missing that would better put the results into context. For example, the reason for controversial results in different regions and sites at the stand level have just recently been reviewed (Toraño Caicoya et al., 2024) and concepts have been developed that describe the limits of carrying capacity (Peters et al., 2018) or generally relate biomass and space occupation (Brown et al., 2004), which could be used to explain the findings here.

For a high-ranking publication, I also would expect that the authors offer a potential explanation of the finding that dying trees are relatively small (L142ff). It is indeed a bit surprising that particularly in very tall trees, smaller individuals have more difficulties transporting water into the canopy than larger ones, or didn't profit from some shading that should decrease evaporation demand. I would thus expect to see some suggestions of what impacts could increase the vulnerability relative to larger trees: less rooting depth, less stem water supply, or a more susceptible wood structure? Based on this, it should be possible to indicate the generality of the findings and their dependency from tree dimension and stand structure. For this discussion it would also be helpful to know about the range of heights in the stands which seems not to be given.

Additional specific remarks:

Figure 1a: What are the different lines and what does the scale mean?

Figure 2a: The description of Fig. 1b should be improved. Why are there lines and why are they at 9.5 and 12.5 degrees? Are these the averages from all plots below and above 11 °C? Is 11 °C the overall average?

L55/56: A clear objective missing. It is too imprecise to just state: "Here, we build on recent advances in self-thinning modeling (22) to investigate changes in carrying capacity and their environmental drivers in forests dominated)"

L87: What is AHMI? (only explained in line 240, not at first appearance)

L96ff: I think that also for the US (Van Gunst et al., 2016) and China (Hao et al., 2024) there are studies indicating a change of carrying capacity with time. Perhaps there are more. The concept of changing self-thinning slopes has also been theoretically investigated before, which I think should be mentioned (Herberich et al., 2020; McDowell et al., 2020).

L97ff: It is true, that N is the most likely driver for increased carrying capacity (see also (Hättenschwiler & Körner, 1998)). Nevertheless, I think, that increasing CO₂ should also be mentioned. Although it doesn't seem to increase the aboveground biomass, the belowground carbon content at least is likely to increase (Kubiske et al., 1997).

L238ff: I am a bit puzzled about the selected climate predictors. For the self-thinning line. Why isn't there something that indicates evaporation demand (e.g. vpd, PET, PET/Prec.). Apart from water availability, these are the most important variables defining drought stress.

Mentioned references

Brown JH, Gillooly JF, Allen AP, Savage VM, West GB. 2004. Toward a metabolic theory of ecology. *Ecology* 85(7): 1771-1789.

Hao Y, Chou J, Zhao W, Li Y, Jin H. 2024. Spatial and temporal pattern of forest carrying capacity and its influencing factors in China, Japan, and Korea based on climate change. *Frontiers in Forests and Global Change* 7.

Hättenschwiler S, Körner C. 1998. Biomass allocation and canopy development in spruce model ecosystems under elevated CO₂ and increased N deposition. *Oecologia* 113(1): 104-114.

Herberich MM, Gayler S, Anand M, Tielbörger K. 2020. Biomass–density relationships of plant communities deviate from the self-thinning rule due to age structure and abiotic stress. *Oikos* 129(9): 1393-1403.

Kubiske ME, Pregitzer KS, Mikan CJ, Zak DR, Maziasz JL, Teeri JA. 1997. *Populus tremuloides* photosynthesis and crown architecture in response to elevated CO₂ and soil N availability. *Oecologia* 110(3): 328-336.

McDowell NG, Allen CD, Anderson-Teixeira K, Aukema BH, Bond-Lamberty B, Chini L, Clark JS, Dietze M, Grossiord C, Hanbury-Brown A, et al. 2020. Pervasive shifts in forest dynamics in a changing world. *Science* 368(6494): eaaz9463.

Peters R, Olagoke A, Berger U. 2018. A new mechanistic theory of self-thinning: Adaptive behaviour of plants explains the shape and slope of self-thinning trajectories. *Ecological Modelling* 390: 1-9.

Toraño Caicoya A, Biber P, del Río M, Ruiz-Peinado R, Arcangeli C, Matthews R, Pretzsch H. 2024. Self-thinning of Scots pine across Europe changes with solar radiation, precipitation and temperature but does not show trends in time. *Forest Ecology and Management* 552: 121585.

Van Gunst KJ, Weisberg PJ, Yang J, Fan Y. 2016. Do denser forests have greater risk of tree mortality: A remote sensing analysis of density-dependent forest mortality. *Forest Ecology and Management* 359: 19-32.

(Remarks on code availability)

Version 1:

Reviewer comments:

Reviewer #3

(Remarks to the Author)

I appreciate the modifications of the manuscript, which have answered my questions and improved the significance of the findings. I am intrigued about the hypothesis stated that larger tree mortality might occur predominantly in structured forests where larger trees may belong to more susceptible species. Although some literature references are given that illustrate the finding that larger trees are more susceptible, it is not clear which of these studies are really supporting the species hypothesis. So please give appropriate references.

I might not have a full overview in this topic but I can see that the hypothesis is e.g. supported by Rowland et al. 2015. Also finding from Fettig et al. 2019 might be interpreted this way. I also think it is necessary to include the discussion From Stephenson and Das 2020, resp. Stovall et al. 2020, which followed the Stovall et al. 2019 publication. In addition, it might be a good idea to expand this hypothesis of mixed forests into structured forests in general, which would not only account for species differences but also for the fact that larger trees should be older and thus may have experienced destructive stress before.

Stephenson, N. L., and Das, A. J.: Height-related changes in forest composition explain increasing tree mortality with height during an extreme drought, *Nature Commun.*, 11, 3402, <https://doi.org/10.1038/s41467-020-17213-5>, 2020.

Stovall, A. E. L., Shugart, H. H., and Yang, X.: Reply to "Height-related changes in forest composition explain increasing tree mortality with height during an extreme drought", *Nature Commun.*, 11, 3401, <https://doi.org/10.1038/s41467-020-17214-4>, 2020.

(Remarks on code availability)

REVIEWER COMMENTS

Reviewer #1 (Remarks to the Author):

This manuscript examines how spatial and temporal differences in climate influence the mortality of *Eucalyptus regnans* forests in Australia. Self-thinning models are fit to tree measurements from plots with many repeated measurements. Importantly, the approach quantifies the effects of climate. An understanding of mortality dynamics is critical for understanding the influence of climate change on forest carbon stocks, and this study shows how such information can be obtained. The manuscript is therefore very interesting and is likely to be of great value to many in this field. It is also well written and I did not note any further suggestions or comments regarding the main text or supplementary information.

Reviewer #1 (Remarks on code availability):

When I ran the R code, I had an error message “Error: Please install the ‘cmdstanr’ package.” But when I tried to install it, I had an error message “Error in FUN(X[[i]], ...) : there is no package called ‘cmdstanr’”.

We appreciate Reviewer #1's positive feedback and supportive assessment of our manuscript.

Regarding the code execution issue, we have added a link to the installation instructions in our code:

```
“library(cmdstanr) #Install instructions: https://mc-stan.org/cmdstanr/articles/cmdstanr.html”
```

The cmdstanr package depends on cmdstan (the command line interface to Stan), which requires separate installation, but following the provided link should make the setup process straightforward.

Reviewer #2 (Remarks to the Author):

The manuscript “Global warming reduces carrying capacity for tallest angiosperm” investigated how drought and heat waves have impacted the carrying capacity of mountain ash forests (*Eucalyptus regnans*), by analyzing silvicultural data from 1947 to 2000 in southeastern Australia. Authors found that warmer conditions and projected temperature increases could reduce tree density and carbon stocks by 24%, with significant implications for forest conservation and global carbon sequestration efforts.

The paper links self-thinning and mortality rates to predict potential carrying capacity dynamics under climate change, providing critical implications for the restoration and management of global forests for carbon sequestration. It is clearly written and includes simple, direct figures, making it an enjoyable read. In my personal view, this manuscript is already very strong. However, I have one suggestion: could the authors make the paragraphs more concise by avoiding the use of single-sentence or two-sentence paragraphs?

We thank reviewer #2 for their careful reading and comments on our manuscript. Following the reviewer's suggestion, we have restructured the text to avoid single-sentence or two-sentence paragraphs, while lightly editing for flow and clarity. For example, in the results section, we had two paragraphs (one on observed impacts of temperature on mortality and carrying capacity, and one on projected carrying capacity in 2080). Similar revisions were made throughout the manuscript where needed.

Original text:

“Carrying capacity was lowest for stands growing in the warmest conditions and further decreased with rising temperatures within each plot. On average, each additional one degree C increase in temperature was associated with a 24.6% increase in mortality rates and a 9% reduction in the carrying capacity of mountain ash stands (Fig. 3 and Table 1).”

And

“The model predicts a three degree C increase in MAT will result in a 24% decline in tree density and carbon content of these forests. The CSIRO RCP8.5 scenario predicts that this region will experience those temperatures by approximately 2080.”

We merged these into a single cohesive paragraph:

“Carrying capacity was lowest for stands growing in the warmest conditions and further decreased with rising temperatures within each plot. On average, each additional one degree C increase in temperature was associated with a 24.6% increase in mortality rates and a 9% reduction in the carrying capacity of mountain ash stands (Fig. 3 and Table 1). Based on these relationships, the model predicts that a three degree C increase in MAT will result in a 24% decline in tree density and carbon content in these forests. The CSIRO RCP8.5 scenario projects this region will experience those temperatures by approximately 2080.”

Reviewer #3 (Remarks to the Author):

The study investigates the stand density/ tree number relationship for Australian trees that potentially grow very high. The authors find that tree mortality is higher and stand density is smaller at warmer sites which are assumed to be dryer too. These results are used to interpolate the biomass carrying capacity into a future that is 3oC warmer. Although the study findings originate from a specific region, I think that they may likely be of general interest and applicable to similar systems. The results are clearly presented and the message is interesting, although neither globally valid nor unique. It is a pleasure to read.

We thank reviewer #3 for his feedback and comments which helped improve our manuscript.

I have, however, some concerns about the discussion of and conclusions from results. Although most of the relevant literature is considered, some papers are clearly missing that would better put the results into context. For example, the reason for controversial results in different regions and sites at the stand level have just recently been reviewed (Toraño Caicoya et al., 2024) and concepts have been developed that describe the limits of carrying capacity (Peters et al., 2018) or generally relate biomass and space occupation (Brown et al., 2004), which could be used to explain the findings here.

We appreciate Reviewer #3's suggestions for additional relevant literature, which we have now incorporated throughout the manuscript.

The Toraño Caicoya et al. (2024) study has been integrated into our discussion of spatial and temporal changes in self-thinning lines:

“We are only aware of four studies exploring temporal variation. Three from Northern Europe and Canada reported increased carrying capacities over time (50; 51; 52), while a fourth study spanning from Spain to Sweden found no significant temporal changes (Toraño Caicoya et al., 2024). This geographical pattern suggests that the trend in carrying capacity may depend on regional climate constraints. In temperature-limited systems, increased carrying capacity may be due to...”

To strengthen our discussion of the mechanistic underpinning of self-thinning processes, we have added several recommended references to the introduction:

“Recent studies have advanced our mechanistic understanding of how tree shape, metabolism, and resource competition drive this self-thinning process (Brown, 2004; Peters, 2018; Herberich, 2020; Mrad, 2020).”

For a high-ranking publication, I also would expect that the authors offer a potential explanation of the finding that dying trees are relatively small (L142ff). It is indeed a bit surprising that particularly in very tall trees, smaller individuals have more difficulties

transporting water into the canopy than larger ones, or didn't profit from some shading that should decrease evaporation demand. I would thus expect to see some suggestions of what impacts could increase the vulnerability relative to larger trees: less rooting depth, less stem water supply, or a more susceptible wood structure? Based on this, it should be possible to indicate the generality of the findings and their dependency from tree dimension and stand structure. For this discussion it would also be helpful to know about the range of heights in the stands which seems not to be given.

We appreciate this important point about size-dependent mortality. To clarify: our findings reflect relative, not absolute, tree size. The dying trees in our study are suppressed individuals that are smaller compared to their neighbors. This pattern is consistent across diverse forest ecosystems, from sessile oak and Douglas fir stands in France to forests in the U.S. and Australia, where dead-to-live tree size ratios typically range from 0.6 to 0.8.

We have expanded our discussion of the physiological mechanisms affecting suppressed trees:

“Despite potentially lower evaporative demand from shading, suppressed trees face multiple physiological limitations: reduced light interception and light use efficiency (Binkley et al. 2010) leads to smaller carbon reserves, while shallower root systems (Le Goff 2001) limit access to deep soil water and nutrients (Pinheiro 2019), resulting in lower water use efficiency (Fernandez 2009).”

We have also clarified an important distinction regarding size-dependent mortality patterns:

“In our monospecific stands, mortality consistently occurred among suppressed trees. In contrast, studies reporting higher mortality among larger and taller trees were often conducted in mixed-species forests, where apparent size-dependent vulnerability may reflect drought-sensitive species occupying upper canopy positions rather than an effect of absolute tree size (i.e., in mixed-forests, species identity can confound tree size)”

Regarding tree heights, we have added dominant height data to Table A1 in the Appendix (range: 10.3 to 73.4 m, mean: 46.1 m). Additional analysis showed no significant relationship between dominant height and the relative size of dying trees. We have added the methodology and results from this analysis to the Materials and Methods section and the Appendix.

Additional specific remarks:

Figure 1a: What are the different lines and what does the scale mean?

We have added the unit to MAT anomalies axis (°C) and expanded the caption of figure 1a for clarity:

“a. Mean annual temperature (MAT) anomalies vs. year for each plot. Colored lines represent individual plots, with colors indicating each plot's average MAT over the 1960-2000 period. Anomalies show the difference between each year's temperature and the plot's mean temperature over 1960-2000, illustrating the temporal warming trends.”

Figure 2a: The description of Fig. 1b should be improved. Why are there lines and why are they at 9.5 and 12.5 degrees? Are these the averages from all plots below and above 11 °C? Is 11 °C the overall average?

We have improved the figure description for clarity. The two lines show model predictions for stands at 9.5°C and 12.5°C, chosen to represent the range of MAT values in our dataset and to demonstrate the expected effect of a 3°C warming by 2080 (CSIRO forecasts). Data points are colored based on their plot's MAT relative to the dataset's mean temperature (11.1°C, rounded to 11°C for visualization). This visualization reveals how plots with above-mean temperatures (red dots) consistently show lower carrying capacity than those below (blue dots).

The updated caption now reads:

“Figure 2: The influence of mean annual temperature ($MAT_{1960-2000}$) on self-thinning allometry. a. Effect of $MAT_{1960-2000}$ on maximum stand stocking per ha. Model predictions from Eq.1 are shown at 9.5°C (blue line) and 12.5°C (red line), representing the approximate temperature range in our dataset and demonstrating the expected impact of a 3°C warming. Data points are colored by whether their $MAT_{1960-2000}$ is below (blue) or above (red) the dataset's mean temperature of 11°C. b. Effect of $MAT_{1960-2000}$ on maximum stand stocking at $D = 25\text{cm}$ (i.e., stand density index) showing a 9% decrease in carrying capacity for each additional °C increase in temperature. Each dot represents one plot measurement. The shaded areas are 95% credible intervals”

L55/56: A clear objective missing. It is too imprecise to just state: “Here, we build on recent advances in self-thinning modeling (22) to investigate changes in carrying capacity and their environmental drivers in forests dominated)”

Thanks for this feedback. We edited the end of the introduction to better streamline the text which now reads:

“Here, we build on recent advances in self-thinning modeling (Trouve, 2017) to investigate changes in carrying capacity and their climate drivers in forests dominated

by mountain ash (Eucalyptus regnans), the tallest flowering plant on Earth and one of the most carbon-dense forests (Keith, 2009; Tng, 2012; Mifsud, 2012). Using long term data spanning five decades (1947-2000), we quantify temporal changes in forest carrying capacity, their associations with climate conditions, and shifts in the relative size of dying trees and their potential climate drivers. Based on these findings, we project future changes in forest carbon stocks under climate warming scenarios."

L87: What is AHMI? (only explained in line 240, not at first appearance).

AHMI is the annual heat moisture index, a measure that combines temperature and precipitation to indicate aridity. It has proven effective for modeling climate impacts on tree growth in Australia (Nitschke, 2017) which is why it was included in our study. We now spelled out AHMI at its first appearance in the text.

L96ff: I think that also for the US (Van Gunst et al., 2016) and China (Hao et al., 2024) there are studies indicating a change of carrying capacity with time. Perhaps there are more. The concept of changing self-thinning slopes has also been theoretically investigated before, which I think should be mentioned (Herberich et al., 2020; McDowell et al., 2020).

We appreciate these reference suggestions and have carefully evaluated each one:

We did not include Van Gunst et al. (2016) because although they found similar patterns of increased mortality in denser stands and during drought, their study relied on remote sensing (Landsat) while ours used field measurements. Given that our discussion now includes ten field studies documenting spatial variation in self-thinning, we chose to maintain methodological consistency by focusing on studies with similar measurement approaches.

We did not include Hao et al. (2024) as it uses a different definition of carrying capacity and focuses on socioecological factors rather than climatic conditions, placing it outside our study's scope.

We have integrated Herberich et al. (2020) into our introduction where we discuss the mechanistic understanding of self-thinning processes:

"Recent studies have advanced our mechanistic understanding of how tree shape, metabolism, and resource competition drive this self-thinning process (Brown, 2004; Peters, 2018; Herberich, 2020; Mrad, 2020)."

We have incorporated McDowell et al. (2020) into our introduction to provide context for broader changes in forest dynamics:

"Despite this threat, few studies have explored how the self-thinning line changes over time in response to climatic variability. This is an important knowledge gap given the widespread changes in forest dynamics (McDowell, 2020) and the need to detect and understand shifts in mortality rates and their drivers (McMahon, 2019; Senf, 2025)."

L97ff: It is true, that N is the most likely driver for increased carrying capacity (see also (Hättenschwiler & Körner, 1998)). Nevertheless, I think, that increasing CO₂ should also be mentioned. Although it doesn't seem to increase the aboveground biomass, the belowground carbon content at least is likely to increase (Kubiske et al., 1997).

We agree that CO₂ fertilization is an important mechanism to consider alongside nitrogen deposition. We have added CO₂ fertilization as a potential driver of increased carrying capacity, citing Walker et al. (2021)'s comprehensive review on the subject.

L238ff: I am a bit puzzled about the selected climate predictors. For the self-thinning line. Why isn't there something that indicates evaporation demand (e.g. vpd, PET, PET/Prec.). Apart from water availability, these are the most important variables defining drought stress.

We appreciate this important point about vapor pressure deficit (VPD) and evaporative demand. We have conducted additional analyses testing VPD's effects and expanded our manuscript accordingly.

Our analysis revealed that both VPD and mean annual temperature (MAT) significantly affected the self-thinning line, with comparable performance in explaining forest carrying capacity (MAT performing marginally better). The strong correlation between VPD and MAT in our study region ($R = 0.96$, $p < 0.001$, see new Figure A2) suggests that temperature effects may be mediated through rising VPD, a known driver of drought-induced plant mortality.

We ultimately focused on MAT for several practical reasons:

1. MAT showed slightly better performance in leave-one-plot-out cross-validation (RMSE of 0.131 versus 0.136 for VPD)
2. VPD data are only available from 1980, while our study includes plots from 1947
3. Reliable VPD projections are not available for future climate scenarios in our region

We have integrated the VPD findings throughout the manuscript.

In the abstract, we now state that:

"forests growing in the warmest and highest vapor pressure deficit conditions had the lowest carrying capacity, and this capacity further decreased with rising temperatures."

In the Materials and Methods section, we have added VPD as a climate predictor alongside MAT, P, and AHMI:

"We considered four climatic predictors that might impact the self-thinning line: mean annual temperature (MAT), annual precipitation (P), vapor pressure deficit (VPD), and the annual heat moisture index (AHMI). HMI combines MAT and precipitation in a simple ratio: $AHMI = (MAT + 10) / (P / 1000)$. This metric, which is the inverse of the

classic Martonne aridity index (84), has proven valuable for modeling tree growth in the region (41). vapor pressure deficit, which measures the difference between actual and saturated air moisture, has been linked to drought-induced plant mortality (58). Before fitting the models, we transformed the climate variables to simplify interpretation: MAT and VPD were centred, while P and AHMI were both centred and scaled. This ensured the intercept reflects the average population response.

Among the four climate predictors (parameters shown in Table A2 in the Appendix), only MAT and VPD had significant effects on the self-thinning line. The MAT model performed best in leave-one-plot-out cross-validation, with an RMSE of 0.131, compared to 0.136 for VPD, 0.154 for precipitation, and 0.152 for AHMI (see Table A3 in Appendix). Given the strong correlation between MAT and VPD in our dataset ($R = 0.96$, see Fig. A2 in Appendix), their similar performance was expected. However, due to VPD data availability (starting only from 1980 versus records extending back to 1900 for the other variables) and the lack of reliable VPD projections for future climate scenarios, we focused subsequent analyses on MAT while acknowledging that its effects may be mediated through increased atmospheric water demand (i.e., rising VPD)."

In the Discussion section, we have edited our text to integrate the effect of VPD, specifically, how warming effects may be mediated through rising VPD:

"While the strong correlation between temperature and VPD in our study region (Fig. A2) makes their individual effects difficult to disentangle, this relationship suggests that warming effects may be mediated through rising VPD, an important driver of drought-induced plant mortality (Eamus, 2013; Grossiord, 2020)."

We have also added new supporting materials: Tables A2 and A3 in the Appendix showing model comparisons, and Figure A2 (see below) illustrating the VPD-MAT correlation:

Figure A2: Relationship between 30-year climate averages (1981-2010) of mean annual vapor pressure deficit and mean annual temperature across 112 forest plots. The solid line shows the linear regression fit ($R = 0.96, p < 0.001$). This figure was added to Appendix.

Mentioned references

Brown JH, Gilgooly JF, Allen AP, Savage VM, West GB. 2004. Toward a metabolic theory of ecology. *Ecology* 85(7): 1771-1789.

Hao Y, Chou J, Zhao W, Li Y, Jin H. 2024. Spatial and temporal pattern of forest carrying capacity and its influencing factors in China, Japan, and Korea based on climate change. *Frontiers in Forests and Global Change* 7.

Hättenschwiler S, Körner C. 1998. Biomass allocation and canopy development in spruce model ecosystems under elevated CO₂ and increased N deposition. *Oecologia* 113(1): 104-114.

Herberich MM, Gayler S, Anand M, Tielbörger K. 2020. Biomass–density relationships of plant communities deviate from the self-thinning rule due to age structure and abiotic stress. *Oikos* 129(9): 1393-1403.

Keith, H., Mackey, B. G. & Lindenmayer, D. B. 2009. Re-evaluation of forest biomass carbon stocks and lessons from the world's most carbon-dense forests. *Proceedings of the National Academy of Sciences* 106, 11635–11640.

Kubiske ME, Pregitzer KS, Mikan CJ, Zak DR, Maziasz JL, Teeri JA. 1997. *Populus tremuloides* photosynthesis and crown architecture in response to elevated CO₂ and soil N availability. *Oecologia* 110(3): 328-336.

Senf et al. International Tree Mortality Network. 2025. Towards a global understanding of tree mortality. *New Phytol* n/a.

McMahon, S. M., Arellano, G. & Davies, S. J. 2019. The importance and challenges of detecting changes in forest mortality rates. *Ecosphere* 10.

McDowell NG, Allen CD, Anderson-Teixeira K, Aukema BH, Bond-Lamberty B, Chini L, Clark JS, Dietze M, Grossiord C, Hanbury-Brown A, et al. 2020. Pervasive shifts in forest dynamics in a changing world. *Science* 368(6494): eaaz9463.

Mifsud, B. et al. 2025. Tasmania's giant eucalypts: discovery, documentation, macroecology and conservation status of the world's largest angiosperms. *Aust. J. Bot.* 73.

Mrad, A. et al. 2020. Recovering the metabolic, self-thinning, and constant final yield rules in mono-specific stands. *Frontiers in Forests and Global Change* 3.

Nitschke, C. R. et al. 2017. The influence of climate and drought on urban tree growth in southeast Australia and the implications for future growth under climate change. *Landscape and Urban Planning* 167, 275-287.

Peters R, Olagoke A, Berger U. 2018. A new mechanistic theory of self-thinning: Adaptive behaviour of plants explains the shape and slope of self-thinning trajectories. *Ecological Modelling* 390: 1-9.

Tng, D. Y. P., Williamson, G. J., Jordan, G. J. & Bowman, D. M. J. S. 2012. Giant eucalypts - globally unique fire-adapted rain-forest trees? *New Phytol* 196, 1001–1014.

Toraño Caicoya A, Biber P, del Río M, Ruiz-Peinado R, Arcangeli C, Matthews R, Pretzsch H. 2024. Self-thinning of Scots pine across Europe changes with solar radiation, precipitation and temperature but does not show trends in time. *Forest Ecology and Management* 552: 121585.

Trouvé, R., Nitschke, C. R., Robinson, A. P. & Baker, P. J. 2017. Estimating the self-thinning line from mortality data. *Forest Ecology and Management* 402, 122-134.

Van Gunst KJ, Weisberg PJ, Yang J, Fan Y. 2016. Do denser forests have greater risk of tree mortality: A remote sensing analysis of density-dependent forest mortality. *Forest Ecology and Management* 359: 19-32.

Walker et al. 2021. Integrating the evidence for a terrestrial carbon sink caused by increasing atmospheric CO₂. *New Phytologist* 229 (5): 2413-2445.

Reviewer #3 (Remarks to the Author):

I appreciate the modifications of the manuscript, which have answered my questions and improved the significance of the findings. I am intrigued about the hypothesis stated that larger tree mortality might occur predominantly in structured forests where larger trees may belong to more susceptible species. Although some literature references are given that illustrate the finding that larger trees are more susceptible, it is not clear which of these studies are really supporting the species hypothesis. So please give appropriate references.

I might not have a full overview in this topic but I can see that the hypothesis is e.g. supported by Rowland et al. 2015. Also finding from Fettig et al. 2019 might be interpreted this way. I also think it is necessary to include the discussion from Stephenson and Das 2020, resp. Stovall et al. 2020, which followed the Stovall et al. 2019 publication. In addition, it might be a good idea to expand this hypothesis of mixed forests into structured forests in general, which would not only account for species differences but also for the fact that larger trees should be older and thus may have experienced destructive stress before.

References:

Fettig, C J. et al. 2019. Tree mortality following drought in the central and southern Sierra Nevada, California, U.S. *Forest Ecology and Management*, Vol. 432 p. 164-178. <https://doi.org/10.1016/j.foreco.2018.09.006>

Rowland, L. et al. Death from drought in tropical forests is triggered by hydraulics not carbon starvation. *Nature* 528, 119–122 (2015). <http://dx.doi.org/10.1038/nature15539>.

Stephenson, N. L., and Das, A. J.. 2020. Height-related changes in forest composition explain increasing tree mortality with height during an extreme drought, *Nature Commun.*, 11, 3402, <https://doi.org/10.1038/s41467-020-17213-5>

Stovall, A. E. L., Shugart, H. H., and Yang, X. 2020. Reply to “Height-related changes in forest composition explain increasing tree mortality with height during an extreme drought”, *Nature Commun.*, 11, 3401, <https://doi.org/10.1038/s41467-020-17214-4>.

Response: We thank the reviewer for this important suggestion and have revised the discussion paragraph on the size of trees that died to better reflect the debate in this area. The new paragraph now explicitly cites studies that demonstrate how species composition, rather than size per se, can drive apparent size-related mortality patterns. The new paragraph reads:

“Trees that died were consistently smaller than their neighbours (i.e., they were suppressed), averaging 62% of their diameter (Fig. 4). This size ratio was stable across all tested climatic gradients. While several studies report greater drought-related mortality among larger trees (Bennett et al., 2015; Rowland et al., 2015; Grote et al., 2016; Stovall et al., 2019), others, including ours, find

higher vulnerability among suppressed individuals (Merian and Lebourgeois, 2011; Grote et al., 2016; Trouve et al., 2017a; Taccoen et al., 2021). Reconciling these patterns requires accounting for relative size within a stand (i.e., social or canopy status), absolute size, and potential confounders such as stand structure and species composition. In mixed-species forests, the apparent vulnerability of large trees can stem from drought-sensitive taxa, particularly fast-growing pioneer species, dominating the upper canopy (Stephenson and Das, 2020), but see Stovall et al. (2020) for an alternative view. Size effects may also be driven or amplified by indirect mortality agents, for example bark beetles which preferentially attack large pines during droughts (Fettig et al., 2019). By contrast, our monospecific, even-aged stands lack these confounders, and mortality concentrates among suppressed trees, consistent with self-thinning dynamics. This vulnerability stems from physiological and morphological constraints: reduced light interception, smaller carbon reserves, and limited access to deep soil water due to shallow root systems (Fernandez and Gyenge, 2009; Binkley et al., 2010; Le Goff and Ottorini, 2001; Pinheiro et al., 2019).”

Note: I used author-year formatting for the citations here to facilitate reading, but in the manuscript I used the usual Nature formatting (numbers) for the citations.